

# Towards EPPS21 nuclear PDFs

Kari J. Eskola[1,2], Petja Paakkinen[3*], Hannu Paukkunen[1,2] and Carlos A. Salgado[3]

**1** University of Jyvaskyla, Department of Physics,
P.O. Box 35, FI-40014 University of Jyvaskyla, Finland
**2** Helsinki Institute of Physics, P.O. Box 64, FI-00014 University of Helsinki, Finland
**3** Instituto Galego de Física de Altas Enerxías (IGFAE),
Universidade de Santiago de Compostela, E-15782 Galicia-Spain

⋆ petja.paakkinen@usc.es

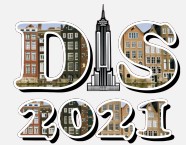

*Proceedings for the XXVIII International Workshop
on Deep-Inelastic Scattering and Related Subjects,
Stony Brook University, New York, USA, 12-16 April 2021*

## Abstract

We report on the progress in updating our global analysis of nuclear PDFs. In particular, we will discuss the inclusion of double differential 5.02 TeV dijet and D-meson measurements, as well as 8.16 TeV W-production data from p-Pb collisions at the LHC. The new EPPS21 analysis will also involve recent JLab data for deep-inelastic scattering. As a novel aspect within our approach, we now also quantify the impact of free-proton PDF uncertainties on our extraction of nuclear PDFs.

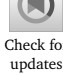
## 1 Introduction

Since our previous EPPS16 [1] global analysis of nuclear parton distribution functions (PDFs), a good number of new data from LHC p-Pb collisions have become available, providing stringent constraints on the gluon content of the lead nucleus [2,3]. It is therefore timely to update our analysis with these new constraints from 5.02 TeV dijet [4] and D-meson [5] and 8.16 TeV W-production [6] measurements included. Here, we present the preliminary results of the work towards our next release of next-to-leading order (NLO) nuclear PDFs with Hessian uncertainties, which we refer to as EPPS21. We discuss also the recent JLab data for deep-inelastic scattering [7], which are now included in the fit. In our analysis, we pay special attention to the role of free-proton PDF uncertainties, and as a new element, we chart their impact on our extraction of nuclear PDFs and provide tools for a general user to propagate this uncertainty into the observables.

## 2 EPPS21 approach to nPDF fitting

As in our earlier analyses [1,8], we define the nPDFs in terms of the nuclear modification factors $R_i^{p/A}$, such that the PDF of a parton $i$ in a proton bound to a nucleus $A$ ($f_i^{p/A}$) can be obtained from that of the free proton ($f_i^p$) as

$$f_i^{p/A}(x,Q^2) = R_i^{p/A}(x,Q^2) f_i^p(x,Q^2),\tag{1}$$

where $x$ is the fraction of nucleon momentum carried by the parton, $Q^2$ is the factorization scale, and for simplicity we take $f_i^{p/A}(x>1,Q^2)=0$. As usual, we parametrize the $x$ dependence of $R_i^{p/A}$ at the scale $Q_0 = m_{\text{charm}} = 1.3$ GeV using a phenomenologically motivated piecewise function, which in this preliminary fit takes the form

$$R_i^{p/A}(x,Q_0^2) = \begin{cases} a_0 + a_1(x - x_a)\left[e^{-x\alpha/x_a} - e^{-\alpha}\right], & x \leq x_a, \\ b_0 x^{b_1}(1-x)^{b_2} e^{x b_3}, & x_a \leq x \leq x_e, \\ c_0 + c_1(c_2 - x)(1-x)^{-\beta}, & x_e \leq x \leq 1, \end{cases}\tag{2}$$

where the flavour dependence is taken impicit on the right hand side. We require continuity and vanishing first derivative at the transition points $x_a, x_e$ (the locations of antishadowing maximum and EMC minimum), whereby the parameters $a_k, b_k, c_k$ can be expressed uniquely as a function of $x_a, x_e$, $y_0 = R_i^{p/A}(x \to 0, Q_0^2)$, $y_a = R_i^{p/A}(x_a, Q_0^2)$, $y_e = R_i^{p/A}(x_e, Q_0^2)$ and the shape parameters $\alpha$ and $\beta$ (after setting $c_0 = 2y_e$).

The $A$ dependence of our fit is parametrized in the $y_r$ ($r = 0, a, e$) with a power-law type function

$$y_r(A) = 1 + \left[y_r(A_{\text{ref}}) - 1\right]\left(\frac{A}{A_{\text{ref}}}\right)^{\gamma_r},\tag{3}$$

where $A_{\text{ref}} = 12$ and the evolution speed is set by the parameters $\gamma_r$. For strange quarks the small-$x$ $A$ dependence is modified by substituting $\gamma_0 \longrightarrow \gamma_0 y_0(A_{\text{ref}})\theta(1 - y_0(A_{\text{ref}}))$ in order to discourage the modification from going strongly negative at the parametrization scale. In addition, the DIS data for lithium were found to suggest somewhat smaller nuclear modifications than what was given by Eq. (3), which is now allowed in the fit by assigning an additional parameter $f_6$ such that

$$R_i^{p/6}(x,Q_0^2) \longrightarrow 1 + f_6\left[R_i^{p/6}(x,Q_0^2) - 1\right]\tag{4}$$

for all flavours. Out of these parameters described above, 24 are kept free in the fit, the rest being either assumed to have the same value across different flavours, set to a physically motivated fixed value, or determined from the sum rules.

Using the bound-proton PDFs obtained with these nuclear modifications, the distributions of a full nucleus with $Z$ protons and $N$ neutrons can be obtained with

$$f_i^A(x,Q^2) = Z f_i^{p/A}(x,Q^2) + N f_i^{n/A}(x,Q^2)\tag{5}$$

by assuming the isospin symmetry between the bound-proton and bound-neutron ($f_i^{n/A}$) distributions. As with the free-proton PDFs, the $Q^2$ evolution of the $f_i^A$ follows the ordinary DGLAP equations, and the scale evolution of the bound-nucleon distributions and nuclear modification factors are deduced from Eqs. (1) and (5). In our analysis, we use two-loop Altarelli–Parisi splitting functions.

Also as in our previous analyses, we include the data as nuclear modification ratios whenever possible to reduce the impact of the free-proton PDFs and other theoretical uncertainties.

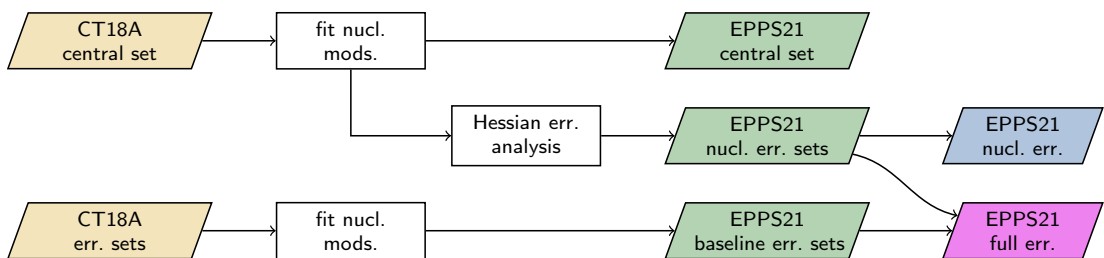

Figure 1: The error analysis flow in the EPPS21 global fit.

This, in addition to using Eq. (1), provides a natural way to decorrelate the $R_i^{p/A}$ from the free-proton PDFs, and to a first approximation our results for the nuclear modifications should be independent of the baseline free-proton PDFs that we use. Some baseline dependence can still persist, and as a new element in our analysis, we study the impact of this additional uncertainty by fitting the nuclear modifications $R_i^{p/A}$ separately for each of the CT18A [9] proton-PDF error sets, the proton-PDF baseline of our analysis, as illustrated in Figure 1. By doing so, we can provide the general user with "baseline error sets", using which one can propagate this additional uncertainty to any desired observable. One can then calculate the full error in the EPPS21 nuclear PDFs by simply adding in quadrature the error from baseline variations with that from the variations of the nuclear modification parameters.

## 3 New data

The main new constraints in our analysis come from the 5.02 TeV dijet [4] and D-meson [5] and 8.16 TeV W-production [6] data from p-Pb collisions at the LHC. As can be seen from Figure 2, we find a very good and consistent fit of all these observables, the only exception being the forwardmost data points (i.e. those with the largest positive rapidity) in the lowest $p_T$ bins of the dijet measurement (see the upper panels of Figure 2). We found the same difficulty in fitting these data points already in a Hessian reweighting analysis [2]. Since the source of this discrepancy is not known, we have excluded these extreme data points from the current fit, but this has little effect on the final results. We note also that the data correlations are not available to us, which might have some impact on the fit quality.

For the $D^0$-production, we employ the S-ACOT-$m_T$ general-mass variable flavour number scheme [11], imposing a $p_T > 3$ GeV cut to reduce the impact of theoretical uncertainties which become relevant in the small-$p_T$ region. We now take systematically into account the luminosity uncertainties, which in the reweighting study [3] were still added in quadrature point by point. As a result, the nuclear-PDF uncertainties remain somewhat larger in the forward direction after the fit.

While for most of the data that we use the nuclear modification ratios are provided directly by the experimental collaborations, in the case of the new W-production measurement at 8.16 TeV [6] we have to construct these ratios by hand. As no proton–proton reference at the exactly same collision energy is available, we use a "mixed-energy ratio", using the CMS proton–proton measurement at 8.0 TeV [10] as a baseline. Although we find a very good fit for these data, they do not seem to give strong additional flavour-separation constraints on top of those that we had already in EPPS16, as we will discuss in the next section. In any case, the good agreement with dijet and D-meson data provides an important check on the nuclear-PDF universality and factorization.

As a new data set in our analysis, we have also included DIS data from the JLab CLAS

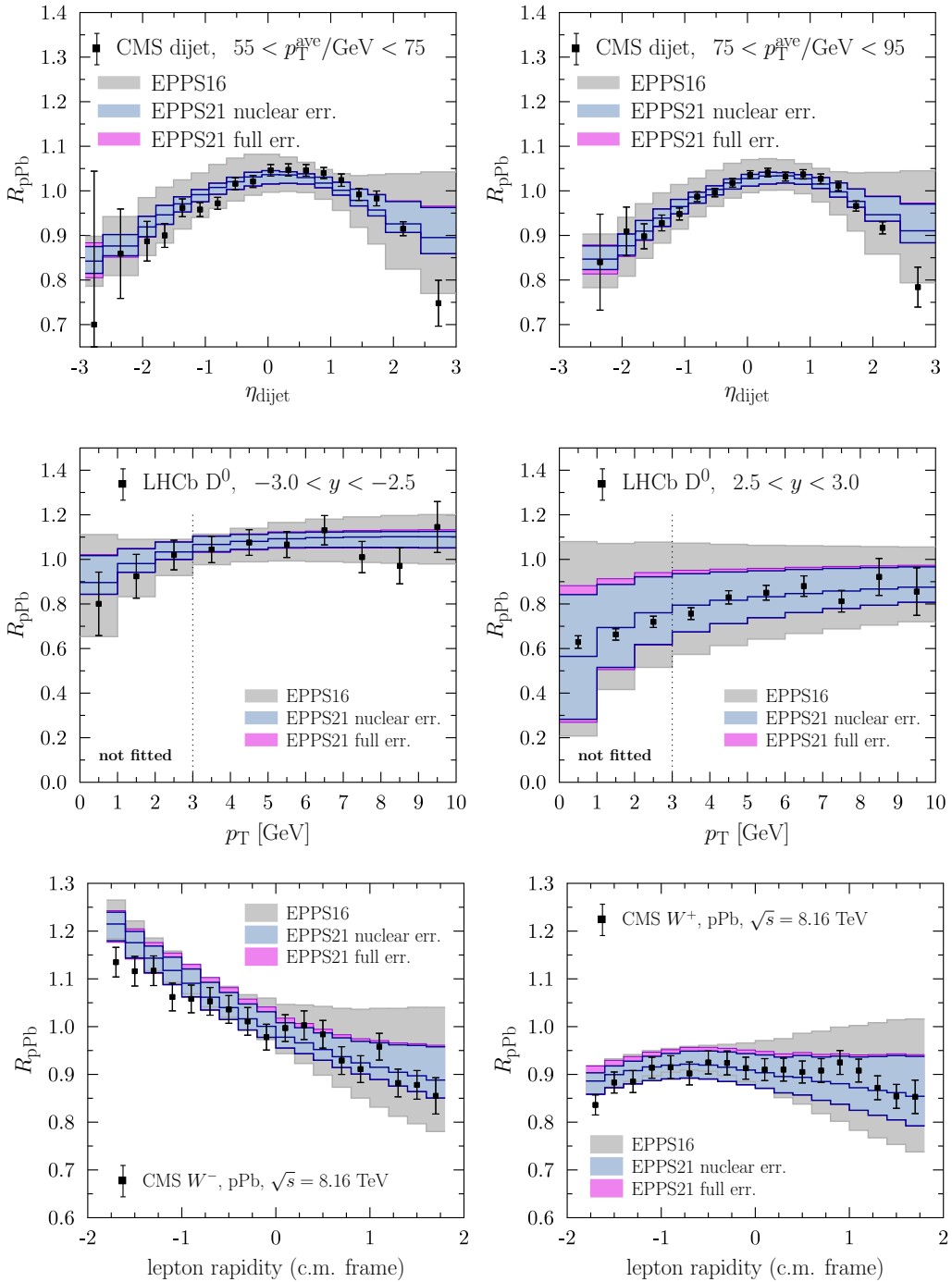

Figure 2: A sample of the new LHC data included in the EPPS21 analysis: The CMS measurement of dijet nuclear modifications [4] in the two lowest $p_T$ bins (top), the LHCb measurement of the nuclear modifications in $D^0$ production [5] in a backward and forward rapidity bin (middle) and the mixed-energy nuclear-modification ratios of the CMS measurements of $W^{\pm}$-production [6,10] (bottom).

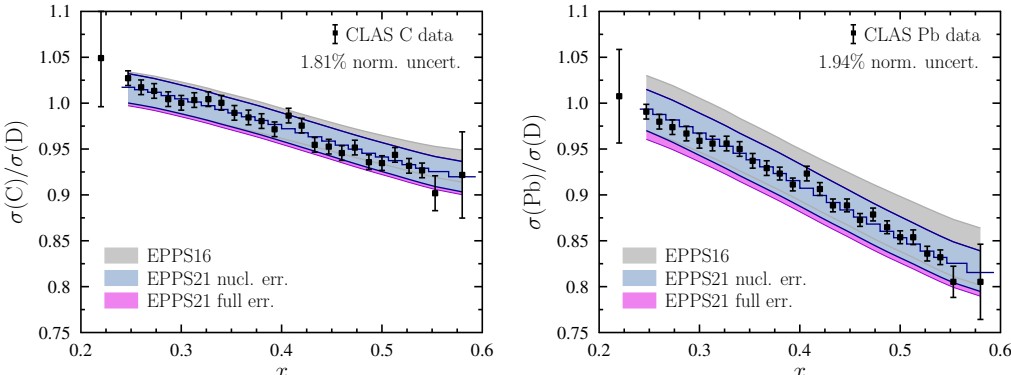

Figure 3: A sample of the new JLab CLAS Collaboration data [7] included in the EPPS21 analysis: The carbon-to-deuteron (left) and lead-to-deuteron (right) ratios.

Collaboration [7], a sample of which is shown in Figure 3. In our calculations, we take into account the leading target-mass corrections. As was shown already in Ref. [12], these data are fully compatible with our framework where the strength of the nuclear modifications depends only on the nuclear mass number $A$. We thus find no need to employ any isospin (i.e. $Z/A$) dependence in our parametrization of the bound-proton nuclear modifications $R_i^{p/A}$, as might be suggested by some EMC-effect models. The impact of these data in the fit is, however, somewhat limited due to normalization uncertainties.

## 4 Results

The resulting nuclear modifications for the lead nucleus from our fit are shown in Figure 4. Instead of $R_i^{p/A}$ defined in Eq. (1), we choose to plot the modifications at a *full nucleus* level,

$$R_i^A(x,Q^2) = \frac{Zf_i^{p/A}(x,Q^2) + Nf_i^{n/A}(x,Q^2)}{Zf_i^p(x,Q^2) + Nf_i^n(x,Q^2)}, \qquad (6)$$

which is more directly connected to what is measured in the observables, taking into account the effect of isospin. The largest impact of the new data is on the gluon modifications, where for the first time in a nPDF global fit we are able to push the uncertainties to a sub-10% level at mid-$x$ and low $Q^2$. These small uncertainties are driven by the very precise dijet data (see Ref. [2]), but also the D$^0$ and $W^\pm$ data lend support for the mid-$x$ gluon enhancement (antishadowing). At small $x$, we have evidence for gluon suppression (shadowing), with the constraints from dijets and $W^\pm$ extending to $x \sim 10^{-3}$ and from the D$^0$ in forward direction all the way to $x \sim 10^{-5}$ (see the discussion in Ref. [3]).

For the quark flavours there are no new strong constraints, and the nuclear-modification uncertainties remain similar to those in EPPS16 or, when taking into account the proton-PDF baseline uncertainty, have even grown from the previous ones. This simply reflects the fact that we do not have enough data to put stringent constraints on the quarks on a flavour by flavour basis. In Figure 4, we show only the results for nuclear modifications in lead, for which we have the strongest new constraints. For lighter nuclei, the nuclear modifications and the associated uncertainties are in general smaller. Exceptions to this rule are, however, the strangeness and gluon content in light nuclei, where we lack direct constraints and the uncertainties in e.g. carbon can even exceed those in lead.

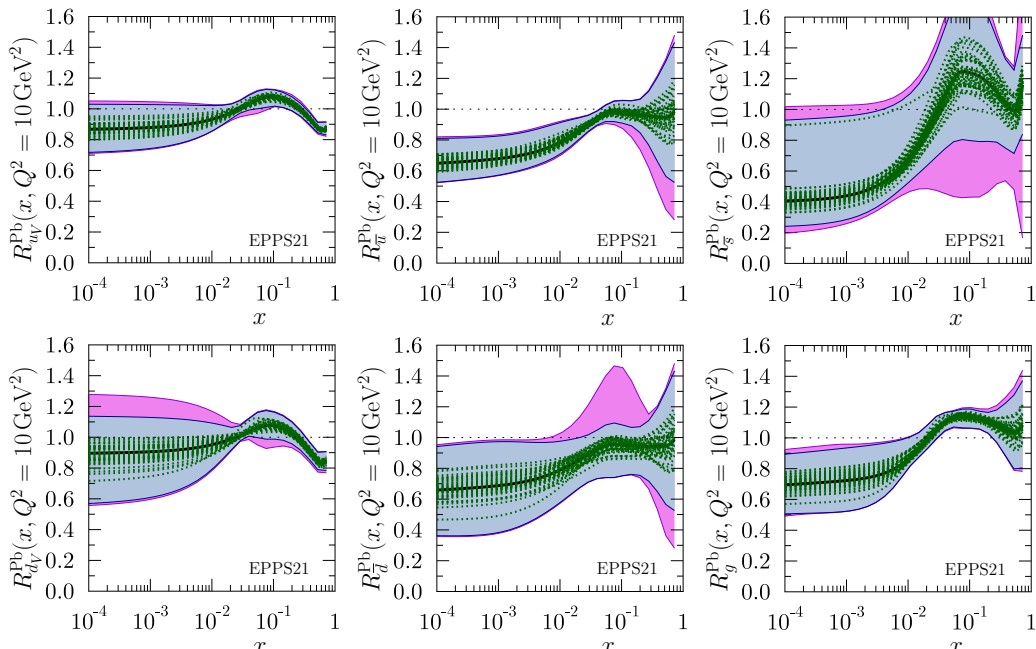

Figure 4: The preliminary EPPS21 nuclear modifications in lead at the 10 GeV$^2$ scale. The blue and purple bands show the nuclear and full errors, respectively, in accordance with Figure 1, and the black solid line and green dotted lines show the results for the central and nuclear error sets, respectively.

## 5 Conclusion

We have presented here our preliminary results for the next generation nuclear PDFs, which will be released under the name EPPS21. We obtain a good fit for multiple new nuclear data from the LHC, but also for new deep-inelastic scattering data from JLab. The LHC data provide strong new constraints on the gluon distribution in lead, where we are able for the first time to push the nuclear modification uncertainties at mid-$x$ and low $Q^2$ to a sub-10% level. On the other hand, the quark flavour separation is difficult to constrain and the uncertainties remain large, especially for the sea quarks.

As a new element in our analysis, we have studied the baseline free-proton PDF sensitivity of the fitted nuclear modifications. We stress that we continue to use data as nuclear modification ratios wherever possible in order to decorrelate the nuclear-modification and free-proton degrees of freedom to best possible extent. Some residual baseline free-proton PDF sensitivity still persists, particularly in those flavour combinations where direct data constraints are scarce, impacting mostly the sea-quark flavour separation, but also light-nuclei gluons.

**Funding information** This work has received financial support from Xunta de Galicia (Centro singular de investigación de Galicia accreditation 2019-2022), by European Union ERDF, and by the "María de Maeztu" Units of Excellence program MDM-2016-0692 and the Spanish Research State Agency (P.P. and C.A.S.). This work was also supported by the Academy of Finland, Projects nr. 297058 and 330448 (K.J.E.), and 308301 (H.P.). The Finnish IT Center for Science (CSC) is acknowledged for the computing time through the Project nr. jyy2580.

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
