# Peer review of "Towards EPPS21 nuclear PDFs"

_SciPost Physics Proceedings, doi:SciPost Phys. Proc. 8, 033 (2022)_

## Round 1 · Referee Report · Anonymous (Referee 1) · 2021-7-19

Strengths

  1. Brief report on nuclear modification fators on PDFs of interest in this subfield
  2. Accounts for the proton PDF uncertainty within a nuclear PDF

Weaknesses

  1. Does not specify all inputs of interest, e.g. order of the fit.

Report

This is a short conference report on ongoing work that will be published fully. It is suitabe for SciPost.
Inevitably it assumes that the reader knows exactly what the authors are talking about. I think a general reader would like to know the order of the resulting PDFs, I expect it is NLO. I also think that the x dependence and A dependence of the nuclear modification factors could be specified in this short paper, rather than left fo the reader to research. Some explanation on the Q^2 dependence of these factors could also be welcome for those used to only proton PDFs.
The work on including the proton PDF uncertainties is very welcome. It appears small, do the authors have any idea if this would remain the case if they had used MSHT20 central PDFs rather than CT18A error PDFs to make this estimate

Requested changes

  1. Specify order of the PDFs 2.Specify functional form of x and A dependence of nuclear modification factors.

---

## Round 1 · Referee Report · Anonymous (Referee 2) · 2021-7-20

Report

These proceedings provide a nice summary of recent work by the EPPS collaboration. It is certainly suitable for publication. I have only one minor suggestion, which is that the meaning of the green lines in Fig. 4 is not I believe made clear at the moment, and would be better clarified.

---

## Round 2 · Author Response

We thank the Referees for their comments and suggestions for improving the manuscript.
* * *
Response to Anonymous Report 1:

The information on the perturbative order of the fit was unintentionally left out from the original manuscript and is now indicated in the text. The functional forms were left unspecified in the original version due to space restrictions and for their preliminary nature. These are now included, with the hope that the Editor will find the increased length of the paper acceptable.

To the Referee's question on whether using MSHT20 instead of CT18A PDFs as a free-proton baseline we can comment that the CT18 uncertainty estimates are moderately conservative and are based on a somewhat smaller data set compared to other contemporary analyses from major fitting groups. It is therefore plausible that using MSHT20 would have resulted in somewhat smaller baseline uncertainty in our fit, but the difference is not likely very large.
* * *
Response to Anonymous Report 2:

The Figure 4 caption has been updated with the information that was missing from the original version.
* * *
With these modifications, we hope that the revised manuscript can now be published in the SciPost Physics Proceedings

Sincerely yours,

Kari J. Eskola, Petja Paakkinen, Hannu Paukkunen, Carlos A. Salgado

---

## Editorial Decision

published